# Evolutionary Multi-Objective Optimization of Extrusion Barrier Screws: Data Mining and Decision Making

**DOI:** 10.3390/polym15092212

**Published:** 2023-05-07

**Authors:** António Gaspar-Cunha, Paulo Costa, Alexandre Delbem, Francisco Monaco, Maria José Ferreira, José Covas

**Affiliations:** 1Institute of Polymers and Composites, University of Minho, 4710-057 Braga, Portugal; 2Institute of Mathematics and Computer Science, University of São Paulo, São Paulo 05508-060, Brazil; 3Portuguese Footwear Research and Technology Centre, 3700-121 São João da Madeira, Portugal

**Keywords:** polymer extrusion, barrier screws, multi-objective optimization, data mining, decision making, number of objectives reduction

## Abstract

Polymer single-screw extrusion is a major industrial processing technique used to obtain plastic products. To assure high outputs, tight dimensional tolerances, and excellent product performance, extruder screws may show different design characteristics. Barrier screws, which contain a second flight in the compression zone, have become quite popular as they promote and stabilize polymer melting. Therefore, it is important to design efficient extruder screws and decide whether a conventional screw will perform the job efficiently, or a barrier screw should be considered instead. This work uses multi-objective evolutionary algorithms to design conventional and barrier screws (Maillefer screws will be studied) with optimized geometry. The processing of two polymers, low-density polyethylene and polypropylene, is analyzed. A methodology based on the use of artificial intelligence (AI) techniques, namely, data mining, decision making, and evolutionary algorithms, is presented and utilized to obtain results with practical significance, based on relevant performance measures (objectives) used in the optimization. For the various case studies selected, Maillefer screws were generally advantageous for processing LDPE, while for PP, the use of both types of screws would be feasible.

## 1. Introduction

Plasticating single-screw extruders are widely used in the industry for the manufacture of an extensive array of plastic products such as pipes, tubing, profiles, film and sheet, electrical wire, filaments, and yarns. In simple terms, an extruder consists essentially of an Archimedes-type screw rotating at a constant frequency inside a hollow heated barrel, which also contains a lateral aperture at one end for feeding the polymer. A shaping die is connected to the opposite end of the barrel. As the screw rotates, the solid polymer depositing on the screw channel inlet is dragged forward, progressively melts, and is forced to flow through the die. As new and more complex polymer systems began to be used (e.g., polymer blends, highly filled polymers, nanocomposites, biodegradable compounds), and more challenging requirements were established (higher outputs, tighter dimensional tolerances, better product consistency), the geometry of extrusion screws evolved considerably. Early screws had a short length-to-diameter ratio (L/D) and contained three geometric zones with different channel depths, commonly denoted as feed, compression, and metering, which are associated with the solids conveying, melting, and melt conveying stages, respectively. However, this simple design evidenced limitations in terms of melting efficiency, dynamic stability, and mixing ability, which progressively led to longer screws, the eventual use of grooved barrels, the insertion of mixing zones in the metering zone, and the development of barrier screws [1,2].

Barrier screws contain a second flight in the compression zone. In this way, they segregate the solid bed from the melt pool during melting, thus improving process stability, and increasing the contact area between the solid bed and the hot metallic channel, thus generating a higher melting rate. It has also been reported that these screws achieve better temperature homogeneity but consume higher specific energy due to the additional stresses developing in the barrier gap [3]. Charles Maillefer [4] patented the first barrier screw (MBS) in 1959. It contains a second flight in the compression zone with a distinct helix angle, connecting the active and passive sides of the main flight. Since then, many other barrier designs have been developed (see, for example, [5,6,7,8,9,10,11]), with many finding practical industrial application in extrusion and/or injection moulding equipment. Rauwendaal [1,2] assessed comparatively various barrier designs, assuming as major performance criteria the evenness of the feed-barrier and barrier-metering geometric transitions, the necessary melting length, the melt conveying capacity, and the manufacturing cost. The author concluded that the MBS is particularly adequate, albeit its relatively limited melting efficiency. The importance of the feed-barrier transition on flow stability was well illustrated by Park and Lyu [12] through the calculation of streamlines and velocity vectors in this region.

Several studies have demonstrated that the performance of barrier screws is rather sensitive to their design features and to the operating conditions selected [11,12,13,14,15]. It is generally presumed that the start of the barrier matches the inception of melting, and that the melting rate is identical to the rate of change of the cross-channel areas for solids and melt in the down-channel direction. However, in practice, a melt film between the barrel and the solid bed must develop upstream of the barrier to avoid plugging. Some designs of the feed-barrier transition are more capable of effectively separating the solid bed from the melt pool. If the barrier gap is too small, the molten film will accumulate as a melt pool in the solids channel, rather than crossing it and depositing in the melt channel. If the barrier is too wide (i.e., if the second flight is too thick), excessive shear heating at high screw speeds may develop.

This work focuses on the design of barrier screws. Given the complexity of the task, purely empirical methods seem to have limited potential. On the other hand, the direct use of numerical modeling routines may turn out to be costly and inefficient, as they rely on the capability of the user to gradually input geometries and/or operating conditions that are more appropriate. Additionally, to the authors’ knowledge, no specific design approaches for this type of screws have been proposed, i.e., where the modeling equations are employed in a prearranged sequence. The authors recently published an extensive review on the optimization of extrusion and other processing techniques [16,17]. These methodologies have been successfully used to design conventional single-screw and co-rotating twin-screw extruders. Consequently, screw design is approached here as a multi-optimization problem, whereby a process modeling package is used judiciously by an optimization algorithm in order to define a Pareto optimal solution. Moreover, a decision-making methodology based on artificial intelligence (AI) techniques is applied to select the best screw.

The paper is organized as follows: Section 2 gives details concerning the modeling of the extrusion process. In Section 3, the methodology used for the optimization is presented in detail, while Section 4 introduces the problem to be solved. The results are presented and discussed in Section 5 and, finally, the conclusions are stated in Section 6.

## 2. Modeling of Plasticizing in Barrier Screws

Several authors modeled melting in barrier screws, often with the aim of creating a tool capable of comparing the performance of various screw types and/or geometries. Ingen Housz and Meijer [9] modified the original Tadmor’s melting model for conventional screws, which stipulates that at any channel cross-section along the melting zone, the solid pellets are compacted and form a continuous bed separated from the inner barrel wall by a thin melt film, which, in turn, feeds a melt pool developing near the active flight of the screw [10]. Considering a similar physical melting model, Amellal and Elbirli [11] numerically solved the various momentum and energy equations coupled to mass balances linking the solid bed, the melt film near to the inner barrel wall, and the melt conveying zones, assuming a non-Newtonian non-isothermal approach and non-uniform solids velocity. Later, the existence of a melt film surrounding the solid bed was also taken into consideration [18]. Han et al. [19,20] extended this analysis by considering the presence of six functional regions at each channel cross-section, namely, solid bed, melt conveying, and four melt films, including at the barrier gap, near the inner barrel surface, near the screw root, and near the screw flights.

Concerning barrier screws for plasticating injection-molding, Kopplmayr et al. [21] took in the co-existence of five different regions, including the melt pool in the solids channel, the melt pool in the melt channel, the melt film between the solids and inner barrel wall, the leakage flow in the gap between the main flight and inner barrel wall, and the melt flow in the barrier gap. A numerical approach based on finite-difference approximation schemes was developed, but little detail was given on the model implementation. With the aim of comparing the performance of various screw profiles, Park and Lyu [12] calculated the pressure, temperature, velocity, and streamlines in barrier screws, assuming non-Newtonian non-isothermal flow and using the Polyflow^®^ commercial software. However, only the presence of melt was taken into consideration, even if the authors recognized—and observed experimentally—that the melt and solids coexist in the solids channel. Huang and Tseng [22] predicted fiber breakage in conventional and barrier screws in injection molding, but again, the melting model adopted was not presented in sufficient detail to understand the underlying physical assumptions.

In general, the above studies assumed that (i) the start of the barrier flight matches the onset of melting, and (ii) the melting rate matches the down-channel rate of change of the cross-channel areas for solids or melt. As discussed in the previous section, experimental evidence has shown the opposite. Therefore, changes either in the operating conditions or in the screw geometry should jeopardize the validity of these assumptions. Consequently, the authors proposed a melting model where the onset and rate of melting were decoupled from the start and position of the barrier [14], which is adopted in the present work. Based on the melting analysis developed by Lindt and Elbirli [23] and Elbirli et al. [24] for a conventional screw, the model takes into account the flow and heat transfer in seven regions of a general barrier screw cross-section, as schematized in Figure 1.

The solids and melt channels are assumed as rectangular and stationary, and the barrel slides at velocity Vb (with components in the transverse, Vbx, and down-channel, Vbz, directions). Region A represents the solid bed, regions C, D, and E identify the melt films surrounding A, regions B and G designate the melt pools in the solids and melt channel, respectively, while F locates the melt film crossing the barrier gap. The melt leakage over the main flight tip is neglected. The flow in the melt films is assumed as one-dimensional, while the flow in the melt pool is taken as two-dimensional. The progress of melting is modeled along sequential down-channel z increments. The mathematical description of each region involves different forms of the momentum and energy equations, together with the relevant boundary conditions, as well as force, heat, and mass balances (see details in [14]).

This melting model was inserted into a global plasticating package that describes the flow and heat transfer along the extruder from the hopper to the die exit by articulately linking the individual process stages developing along the screw through suitable boundary conditions [14]. In the down-channel direction, the stages are the following: (i) gravity conveying of pellets in the hopper; (ii) drag solids conveying induced by friction forces in the first screw turns; (iii) creation and growth of a thin film of melted material separating the solid bed from the surrounding screw wall(s)—this is usually known as the delay zone; (iv) melting in the barrier zone according to the mechanism discussed above; (v) melt conveying; and (vi) melt pressure flow through the die. The location and extent of these stages depend on the local thermo-mechanical conditions, i.e., they are not made a priori coincident with the position of specific geometrical screw features. The plasticating sequence is modeled along successive down-channel and die increments. If the calculated pressure at the die exit is higher than a predefined small value, the calculations are repeated.

Figure 2 illustrates the computational predictions of major process parameters for a Maillefer-type barrier screw (total length L/D = 30, length of the compression (barrier) zone of 5 L/D, compression ratio of 2.5). Figure 2A depicts the axial profiles of the solid bed width (ratio of solids width, X, to solids channel width, Ws_profile_), the average melt temperature, Tmelt, and the viscous dissipation (ratio of Tmelt to the local set temperature, Tb). The beginning of melting does not coincide with the start of the barrier, and the melting rate, X/W, is distinct from the rate of variation of the channel width, Ws_profile_. In the final melting stages (i.e., approximately at L/D = 13), the physical presence of the barrier controls the melting, which will have consequences on the mass output. Figure 2B depicts the axial melt pressure and cumulative mechanical power consumption profiles, demonstrating how the high shear rates developing at the barrier gap originate high shear stresses, and correspondingly, high melt pressures and power consumption [14]. These predictions were globally validated experimentally [25].

## 3. Optimization Methodology

### 3.1. Multi-Objective Optimization

The design of extrusion-barrier screws approached as an optimization problem is multi-objective, meaning that there is the need to satisfy simultaneously several performance measures (objectives), which are often conflicting and can have different importance to the process. A multi-objective optimization problem can be defined mathematically as follows [26,27]:(1)$$\begin{array}{lll} \mathrm{maximise} & f_{m}(x_{i}) & i=1,\ldots ,N;\;m=1,\ldots ,M\\ \mathrm{subject}\;\mathrm{to} & \;\;g_{j}(x_{i})=0 & j=1,\ldots ,J\\ & \;\;h_{k}(x_{i})\geq 0 & k=1,\ldots ,K \end{array}$$
where *M* is the number of objectives.

The various objectives can be dealt with in three ways, as follows: a priori, a posteriori, or iteratively. In the first case, the decision maker (DM) initially defines the relative importance of the objectives and the performance of the solutions can be obtained through the use of aggregation functions, e.g., weighted sum, weighted product, or weighted Tchebycheff metric [28]. The optimum can be found using a traditional single-objective methodology. In the second alternative, the various objectives are optimized simultaneously in order to obtain a set of solutions denoted as the Pareto set. Therefore, two different spaces exist, that of the decision variables and the domain of the objectives. The optimization aims to find the solutions where all objective functions are optimized. The Pareto set is the set of non-dominated solutions, which are the solutions that are incomparable to each other, as it is not possible to state that one is better than another for all objectives simultaneously [26,27]. Figure 3 illustrates this concept using a problem with two objectives to maximize. The comparison between solutions A and B shows that A is better in both objectives and, thus, A dominates B. The same happens when comparing solutions B and C. In this case, C dominates A, which means that all solutions belonging to the dark grey square dominate solution A. However, when comparing any solution within the light grey squares (solutions D and E versus solution A), it is not possible to conclude that one is better than the other. The best solution can be selected by using additional preference information provided a posteriori by the DM [28]. In the third approach, the optimization and selection of solutions can be made iteratively and interleaved, i.e., the optimization algorithm delivers different solutions to the decision maker, who indicates the preferences, and then the optimization algorithm runs again [29].

Taking into account the need to find a set of non-dominated solutions, the best way to deal with multi-objective optimization is to use population-based algorithms, such as evolutionary algorithms (EAs) [26,27]. This type of algorithms is based on the metaphor of natural evolution, i.e., they use the concepts of mutation and crossover to evolve a population of individuals, which contains the potential solutions to the problem under study, along the successive generations. A better opportunity for reproduction is given to individuals with higher performance, that is, to individuals that have more capacity to survive. The new solutions (the offspring) are generated through genetic operators such as crossover and mutation, inheriting most of the parent characteristics. The selection operators enable the best individuals to have a higher probability of being selected for producing offspring, and the variation operators allow for the generation of new individuals [26,27]. Selection is based on the quality of each individual, which is given by a fitness function that is associated with the objective or objective functions for single or multi-objective optimization, respectively.

Recently, several efficient multi-objective optimization EAs (MOEAs) have been proposed [30,31,32,33,34]. Algorithm 1 outlines the general framework of MOEA.
**Algorithm 1** MOEA1: P ← Initialization ()2: Repeat3: R ← MatingSelection (P)4: Q ← Variation (R)5: P ← EnvironmentalSelection (P ∪ Q)6: Until stopping criterion is met

The algorithm starts with the initialization, in which a population of solutions is randomly generated within the search space. Then, this population is subjected to the evolutionary process by the sequential application of the mating selection, variation, and environmental selection procedures. The mating selection consists of selecting parents from the population for reproduction with a higher probability for fitter individuals, considering the existence of multiple objectives. Variation involves the use of evolutionary operators that are applied to the chromosomes of parent individuals for producing offspring. Finally, the environmental selection procedure is based on the concept of the survival of the fittest from natural evolution, forming the population of the next generation from the multiset of the current and offspring populations. The differences between the existing MOEAs are related to the design of its procedures, but all fit in this algorithm.

The present work adopts the S-Metric Selection Evolutionary Multiobjective Optimization Algorithm (SMS-EMOA), proposed by Beume et al. [34], which is a state-of-the-art MOEA that proved to be very efficient in solving real-world optimization problems. In this algorithm, the population is subjected to a steady-state evolutionary process where a single offspring is produced in each generation. The mating selection is performed by selecting a set of different population members at random, but distributed along the whole Pareto front. In the variation operator, a single offspring is generated by recombining selected parent individuals. Then, the environmental selection procedure consists of updating the population by removing the individual with the smallest hypervolume contribution in the last nondominated front. Evolution takes place for the number of generations specified by the user.

### 3.2. Data Mining Methodology and Decision Making

Usually, dealing with Multi-Objective Optimization Problems (MOOP) requires some degree of interaction with a DM, i.e., with an expert in the field. Simultaneously, albeit the high amount of data and the potentially complex relationships between the variables and the objectives and between the objectives, these must be established to support the optimization and decision processes. Moreover, in such complex processes, the DM must understand the optimization process as well as the procedure for selecting solutions. Therefore, it is desirable to (i) use data analysis to define the above relationships; (ii) to determine whether all objectives are really necessary or if their number can be reduced; (iii) to explain the solutions found; and (iv) to provide a very good approximation to the final solution to be used in the real problem studied. This can be performed by linking data analysis tools with optimization methodologies.

For these purposes, this work uses the DAMICORE (proposed in 2011) framework based on the estimation of distances by compression algorithms, called NCD. This algorithm showed a good performance when tested in a problem similar to the one being studied here, simultaneously allowing an easy explanation of the decision process to the DM. In this framework, a Feature Sensitivity Optimization based on Phylogram Analysis (FS-OPA) [35,36,37] is applied to find the set of the principal features of the problem considering the real context, i.e., its feature interactions and their contribution to a target or an objective. In the context of this work, a phylogram is a diagram representing the relationships and distances between different groups of variables and/or objectives in the form of a branching diagram. The branches on a phylogram are proportional to the number of changes between different variables and/or objectives.

DAMICORE is a constructor of models based on phylograms able to deal with any type of data (integer, real and complex numbers, categorical, images, sound, etc., and mixtures of them), sequentially involving the following main tasks:

(1) Generate a distance matrix from the data using the Normalized Compression Distance (NCD) metric [38];

(2) Construct evolutionary trees using phylogram-based modeling. DAMICORE uses a distance reconstruction algorithm called Neighbour Joining (NJ) in which the quality of the models is improved using a systematic resampling strategy [36];

(3) Perform community detection by analyzing the phylograms found and extract important and trustworthy information from them. In this case, a complex network approach known as Fast Newman (FS) is applied [39]. In this way, it will be possible to find subgroups of data that share information (DNA), designated as clades, which identify the communities.

The application of this methodology encompasses the generation of phylograms with information delivered in two levels of learning as follows:In the first level, the aim is to find clades, each representing a cluster of variables sharing information. For optimization purposes, each of these clusters represents the set of variables with important interactions. The result is a table with a list of variables with a cluster per row;In the second level of learning, the FS-OPA calculates the contribution of each clade to the objectives, by measuring the distance between the clades of objectives (oclade) and each variable clade (vclade) using the phylogram obtained. These distances correspond to an estimation of the influence of a clade on the improvement of an objective. As a result, three matrices are produced, one with the phylogram distances from vclades to oclades, the second with the relative phylograms distances from each variable to each objective, and the third with the distances between each objective and the other objectives.

This methodology was recently applied to another extrusion problem with different objectives including (i) to learn from computational data [40], (ii) decision making [41], and (iii) reduction in the number of objectives [42,43]. In the last case, the objectives to be selected were chosen using both the phylogram and the table with the distance of objectives–objectives, as follows:Choose the objective(s) of the less distant clusters;Choose one objective of the more distant (single) cluster;Choose the objective(s) from each of the remaining clusters taking into account the phylogram and the expertise of the DM(s) on the process.

Finally, the WSFM technique [44] was used to quantify the significance of the solutions, taking into account the relative importance of the different objectives. WSFM is based on the concept of rubbers, whereby each point in the Pareto front is connected by a line between the point identifying the solution and perpendicular to the Cartesian axes. Then, a stress (*σ_wi_*) is calculated for each line, taking into account the weights previously defined (*wi*) and the value of the objective at that point. The stress function, *T*(*x*), is determined by the following:(2)$$T(x)=\overset{}{\underset{i}{\max}}\left(\sigma_{wi}\left(f\left(X\right)\right)\right)$$

Finally, the best solutions are those that minimize *T*(*x*) (for more details, see Ferreira et al. [44]).

## 4. The Barrier Screw to Be Designed

The methodology proposed will be applied to the extruder shown in Figure 4, fitted with an MBS. The figure illustrates the main geometrical features of the equipment and identifies the decision variables, which comprise the following: (i) the operating conditions: screw speed (N), barrel temperature profile in three zones (Tb1, Tb2 and Tb3), and (ii) the screw geometry: length of the feed and compression zones (L1 and L2), and internal screw diameters in the feed and metering zones (D1 and D3). In the particular case of an MBS, L2 is the axial length of the barrier, and two additional design variables are also defined, namely, the thickness of the barrier flight (Wf) and the clearance/gap between the barrel and the barrier flight (Hf). The length of the metering zone (L3) is obviously determined from the difference between the total screw length (L) and L1 + L2.

Table 1 presents the physical, thermal, and rheological properties of the two commercial polymers considered for the design of the screw, a low-density polyethylene (LDPE), and a polypropylene (PP), which represent well the typical characteristics of extrusion-grade materials. The rheological properties are quantified by the Carreau-Yasuda law as follows:(3)$$\eta =\eta_{0}\;a_{T}\;{\left(1+{\left(\lambda \;a_{T}\;\dot{\gamma }\right)}^{a}\right)}^{\frac{n-1}{a}}$$
where
(4)$$a_{T}=\exp\left(\frac{E}{R}\;\left(\frac{1}{T}-\frac{1}{T_{0}}\right)\right)$$

In these equations, *η* is the melt viscosity at temperature *T* and at shear rate $\dot{\gamma }$, *η*_0_, *n*, *λ*, and *a* are material constants resulting from adjusting the equation to the experimental data, and *T*_0_ is the reference temperature.

As identified in Table 2, the design/optimization was performed considering six objectives, which consist of major process performance criteria, namely, mass output, length of screw required for melting, average melt temperature at die exit, mechanical power consumption, a measure of distributive mixing (WATS) proposed by Pinto and Tadmor [45], and viscous dissipation. The aim of the optimization (two objectives to maximize and four to minimize) and the allowable range of variation are also specified. Table 3 and Table 4 present the different case studies selected for each material. In Cases 1 to 6 for LDPE, and Cases 8 to 11 for PP, the operating conditions were defined and kept constant, while for Cases 7 and 12, the operating conditions can vary within the range specified. Table 5 indicates the geometrical parameters to be optimized and their range of variation. The optimization exercise was extended to an equivalent conventional three-zone screw (CS), in order to compare the performance of the two types of geometries. An additional variable designated as ‘case’ and ranging in the interval [0,1] was created for this purpose. If ‘case’ is lower than or equal to 0.5, the program activates the variables corresponding to the conventional screw (CS); otherwise, the MBS is considered. In this way, during the evolution process, the EA does not lose information concerning both screws, even if one type of screw is activated for a certain solution.

## 5. Results and Discussion

### 5.1. Optimization Analysis

This section presents and discusses the optimization results using the SMS-EMOA algorithm. Figure 5 displays the bi-objective optimization runs for LDPE, Cases 1 and 4, in terms of length for melting (L) versus output (Q) (Figure 5A,C), and degree of mixing (WATS) versus output (Q) (Figure 5B,D). Due to the random generation of the initial population, the number of CSs and MBSs is very similar. However, when comparing the initial and final populations, it becomes evident that the MOEA used is able to considerably increase the performance of the solutions, since the values for the two objectives are much higher, following the direction shown of the arrows. For Case 1 (Figure 5A,B), for N = 40 rpm and Tbi = 140 °C (Table 3), the best screw to use is the MBS. This is not true for Case 4 (Figure 5C,D), for N = 40 rpm and Tbi = 180 °C (Table 3), where for L vs. Q there is a single solution representing a CS, but for WATS vs. Q there are two regions, one represented by a CS for higher values of the degree of mixing, and another represented by an MBS for higher values of output. This can be explained by the fact that for higher barrel temperatures (Case 4), the polymer melts earlier in the CS due to the heat conducted from the barrel, thus making available a longer channel length for melt conveying and, consequently, for higher WATS. Simultaneously, in this case, the length of the Pareto front is higher, providing more options for choosing the best screw.

Figure 6 compares the results obtained for Cases 1 to 3 in the same domains as before, i.e., L vs. Q and WATS vs. Q. As anticipated, when the screw speed increases (from Case 1 to Case 3), the Pareto optimal front moves in the direction of higher outputs, but in all cases, the best solutions are only for the MBS. However, when the same variation of screw speed is tested together with higher barrel temperature profiles (Cases 4 to 6, Figure 7), the outcome might be different. For the lower screw speed, when optimizing together Q and WATS, the solutions obtained include both CSs and MBSs, again, because of earlier polymer melting in the CS.

Figure 8 shows the results for Case 7, in which the operating conditions (N and Tbi) are also decision variables, i.e., they are allowed to change in the range of variation indicated in Table 3. The results are very similar to those of Case 6, when N and Tbi are fixed at 80 rpm and 180 °C. Indeed, both the screw speed and the barrel temperature profile converge to the upper limits (80 rpm and 180 °C). In fact, Figure 6 demonstrates that the optimal Pareto fronts obtained for Case 3 dominate those for Cases 1 and 2, this being the reason for the similarity of the results for Cases 3 and 7.

Figure 9 and Figure 10 show similar results for PP. Figure 9 displays the initial and final populations for the two bi-objective optimization runs of Case 8, L vs. Q and WATS vs. Q. Again, there is a clear improvement along the generations. However, in this instance, the CS solutions prevail in the final population. This happens because the melting of PP occurs very fast in the screw due to its thermal properties. Consequently, not only it is unnecessary to use a barrier screw to assist/force melting, but the CS also performs better concerning the other optimization objectives, i.e., WATS vs. Q. The same is observed in Figure 10, where the effect of increasing screw speed is depicted. As for LDPE (Figure 6), the optimal Pareto front solutions allowed a higher output when the screw speed was increased. Nevertheless, at higher screw speeds (Figure 10E), the higher outputs are only achieved by MBS, as they imply later polymer melting in the screw.

Figure 11 and Figure 12 present the optimization results for six objectives for LDPE and PP, respectively. Each column depicts the bi-dimensional representation of optimal Pareto fronts for the six objectives for Cases 1, 4, and 7 for LDPE (Figure 11) and for Cases 8, 11, and 12 for PP (Figure 12). Note that this is a six-dimensional space from which it is difficult to infer the best solution (or solutions) to select, since in this high-dimensional space, it is very difficult to find the solution that fits all objectives simultaneously. Therefore, the methodology discussed in Section 3.2 will be applied to Cases 7 and 11. This includes the definition/selection of the relevant objectives using the DAMICORE, and the application of a decision-making methodology to select the best solutions. Throughout this process, it is important that the decision maker has a good understanding about the solutions found. This will be performed in the next section.

### 5.2. Analysis of the Optimization Process

The application of DAMICORE to the six objectives problem of Case 7 yields the phylograms for the initial and final populations of Figure 13 and the resulting Tables of distances 6 and 7. In the figure, the objectives are identified in boxes. The decision variables (DVs) and the objectives are clustered, taking into account the NCD metrics (Section 3.2). The observation of the phylograms allows for the identification of the clusters that share information, as well as the distances of DVs–DVs, DVs–objectives, and objectives–objectives. The distances are represented by the path that is necessary to go through, i.e., the length of the branches on a phylogram. For the final population, the objectives are grouped in the sets (Q, L), (Power, WATS), and (T) and (TTb). From these phylograms, it is possible to determine those distances. For example, Table 6 shows the distances of DVs–objectives for the final population ordered from the lowest to the highest. The following conclusions can be drawn:On average, the most important DVs are L2_, case, N, and L1_;Q and L are more influenced by L2_ and L1_;Power and WATS are influenced by L2_, case, N, and L1_;T and TTb constitute two separate groups, even considering that TTb is equal to T/Tb, and this happens because Tb is changing.

The question now is how to choose the solution (or solutions) to be used, based on this six-dimensional objective space. The application of any aggregation method would be limited, since the DM would not be informed of any explanation concerning the choice, i.e., they must trust the method. A better way is to check the existing relations between the objectives in order to possibly remove a few from the process of decision. For that purpose, the rules defined at the end of Section 3.2 are applied to Table 7, where the distances between the objectives are presented. The application of Rule 1 allows selecting Power and WATS, Rule 2 allows selecting TTb, and Rule 3 selects Q (or any of the others with the same distance, L or T). Then, by applying the WSFM (Equation (2)) for the objectives selected (Q, Power, WATS, and TTb), the results presented in Table 8 are obtained. In this example, two different sets of weights are used, one attributing equal importance to all objectives (0.25), and the other attributing higher importance to output (0.5) and equal significance to the remaining (0.1667). The best solutions are those with a lower t(X) value. Table 8 demonstrates that these two sets of solutions have a balanced performance when taking into account all objectives. When the relative importance of output is higher, the other decision variables are adjusted to maintain this equilibrium, but different solutions are found in both cases. Additionally, these adjustments are made in screw speed (N), barrel temperature in the third zone (Tb3), length of the feed zone (L1_), height of channel in the metering zone (H3_), and Pitch (P_). These results are coherent with the practical response of the extruder, i.e., by following this decision process, the DM is able to understand the optimization mechanism and is also able to select an informed solution.

When applied to PP, the DM process is more complex because the final Pareto optimal fronts include both the CS and MBS. Indeed, the most relevant DVs indicated in Table 9 (obtained from the phylogram of Figure 14 are related to both types of screws, namely, ‘case’, L2_, L1_, N, L2, and L1. Following the same strategy, i.e., based on the distances of objectives–objectives (Table 10) and on the rules of Section 3.2, Q, T, Power, WATS, and TTb are selected. In this situation, only objective L could be discarded.

Table 11 shows the solutions chosen for three different sets of weights as follows: (i) objectives with identical importance (weights equal to 0.2), (ii) output with higher importance (weight equal to 0.4, the remaining equal to 0.1), and iii) output with predominant importance (weight equal to 0.6, the remaining equal to 0.08). The results are the same for the sets of weights (i) and (ii), including two MBSs and three CSs, while for set (iii), all screws are MBSs. In all solutions found, the operating conditions do not change. When a CS was selected, the balance between the solutions was accomplished at the cost of L and P, while for the MBS, the relevant DVs are only L1_ and L2_. However, the range of variation of the objectives for these optimal Pareto front solutions is higher. This is probably due to the presence of more objectives in the calculation of t(X), when compared to the results obtained for LDPE.

This demonstrates that in a complex real-world optimization problem such as barrier extrusion screw design, it is important to offer the DM not only the solutions, but also an insight about the problem.

## 6. Conclusions

Artificial intelligence techniques, namely, data mining, decision making, and multi-objective evolutionary algorithms, were applied to design Maillefer-barrier screws, simultaneously considering the influence of major process parameters and the processing of two polymers commonly used in industrial extrusion, LDPE and PP. The competing performance of conventional screws was also simultaneously taken into consideration.

The optimization methodology adopted was sensitive to the different thermophysical characteristics of the polymers. Barrier screws are advantageous for processing LDPE, except for low screw speeds, in which case the two types of screws can be used. For PP, the optimization methodology suggests the use of both types of screws for a wider range of operating conditions. While proposing the most adequate screw geometries, the methodology adopted evidenced the correlations between the process parameters selected for the design, thus keeping the decision maker informed about the reasons for the selection.

The methodology illustrated here should be directly applicable to other polymer processing optimization problems.

## Figures and Tables

**Figure 1 polymers-15-02212-f001:**
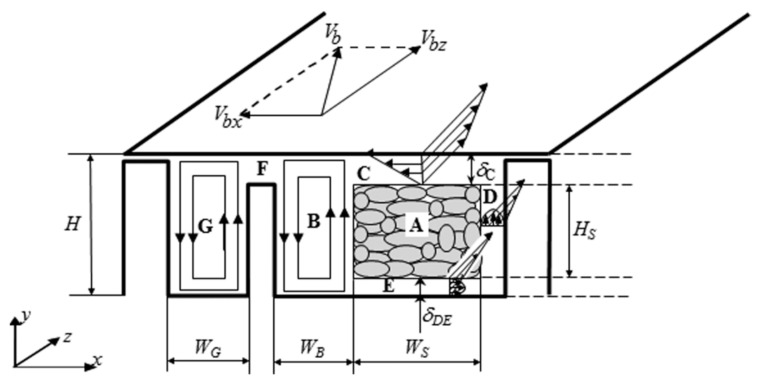
Melting mechanism in the cross-section of a barrier screw. A—solid bed; C, D, E—melt films surrounding the solid bed; B, G—melt pools in the solids channel and melt channel, respectively; F—melt film crossing the barrier gap. Vb is the barrel sliding velocity, W_B_ and W_G_ are the widths of melt pools B and G, respectively, W_S_ is the width of the solid bed, H is channel depth, H_S_ is the depth of the solid bed, and δ denotes the thickness of a melt film (reproduced with permission from [14]).

**Figure 2 polymers-15-02212-f002:**
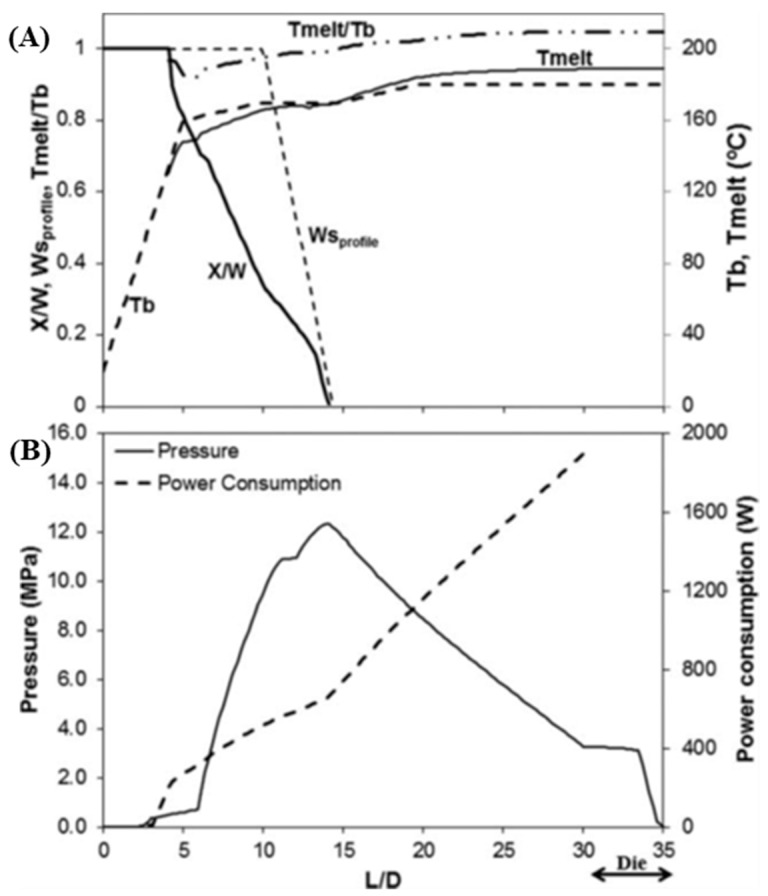
Computational predictions for a Maillefer-type barrier; (**A**) axial profiles of the solid bed width (ratio of solids width, X, to solids channel width, Ws_profile_), average melt temperature, Tmelt, and viscous dissipation (ratio of Tmelt to the local set temperature, Tb); (**B**) axial melt pressure and cumulative mechanical power consumption profile (reproduced with permission from [14]).

**Figure 3 polymers-15-02212-f003:**
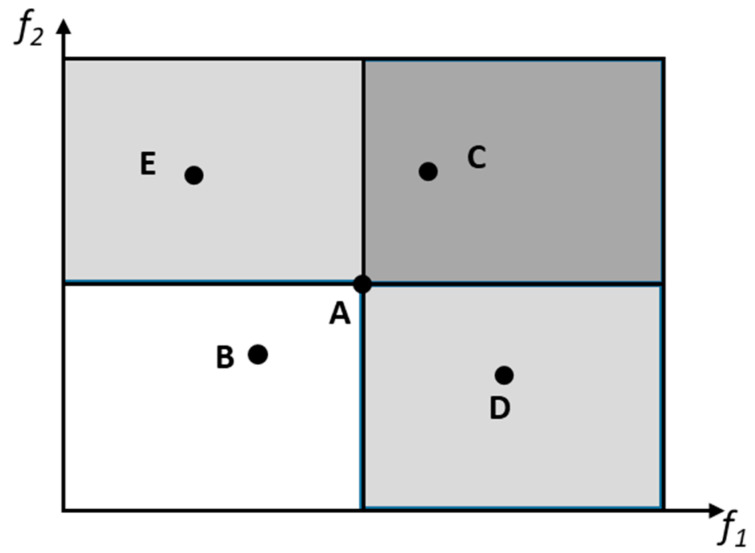
Concept of non-dominance. *f*_1_ and *f*_2_ are two objectives to maximize, A to D are possible solutions.

**Figure 4 polymers-15-02212-f004:**
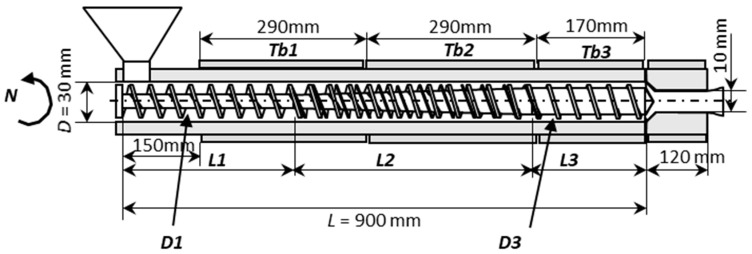
Extruder geometry.

**Figure 5 polymers-15-02212-f005:**
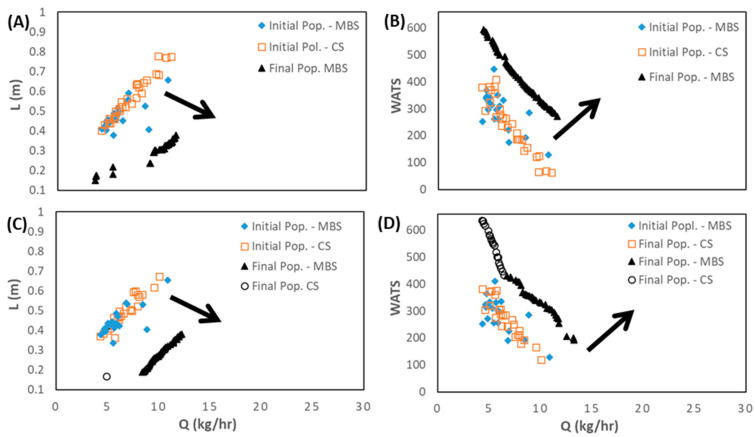
Bi-objective optimization (length for melting, L, and WATS) for LDPE as follows: (**A**) and (**B**) Case 1; (**C**,**D**) Case 4 (the number of solutions in the initial population for each type of screw, CS and MBS, is similar). The arrows indicate the direction of optimization.

**Figure 6 polymers-15-02212-f006:**
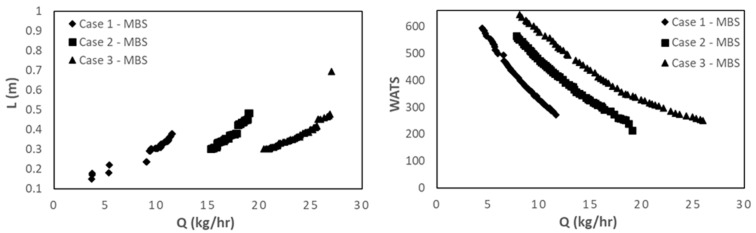
Bi-objective optimization (length for melting, L, and WATS) for LDPE for Cases 1 to 3.

**Figure 7 polymers-15-02212-f007:**
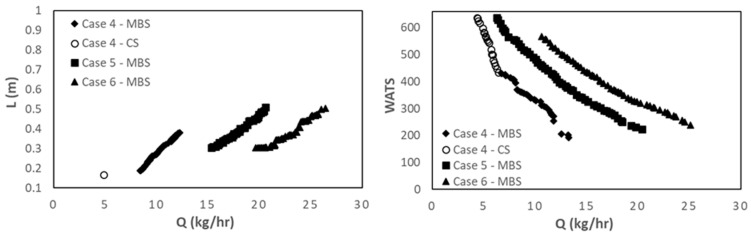
Bi-objective optimization (length for melting, L, and WATS) for LDPE for Cases 4 to 6.

**Figure 8 polymers-15-02212-f008:**
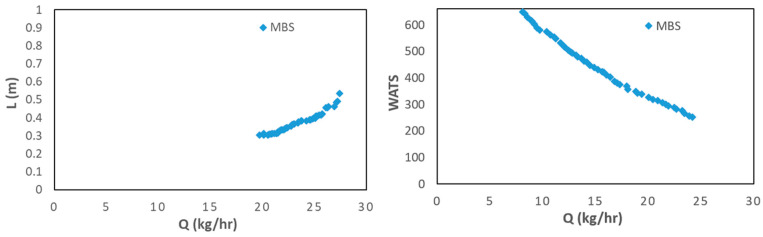
Bi-objective optimization (length for melting, L, and WATS) for LDPE for Case 7 (**left**—Q vs. L; **right**—Q vs. WATS).

**Figure 9 polymers-15-02212-f009:**
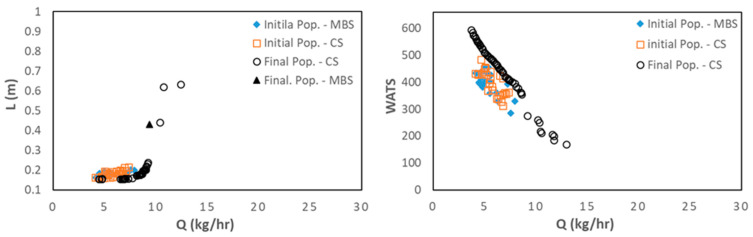
Bi-objective optimization (length for melting, L, and WATS) for PP for Case 8.

**Figure 10 polymers-15-02212-f010:**
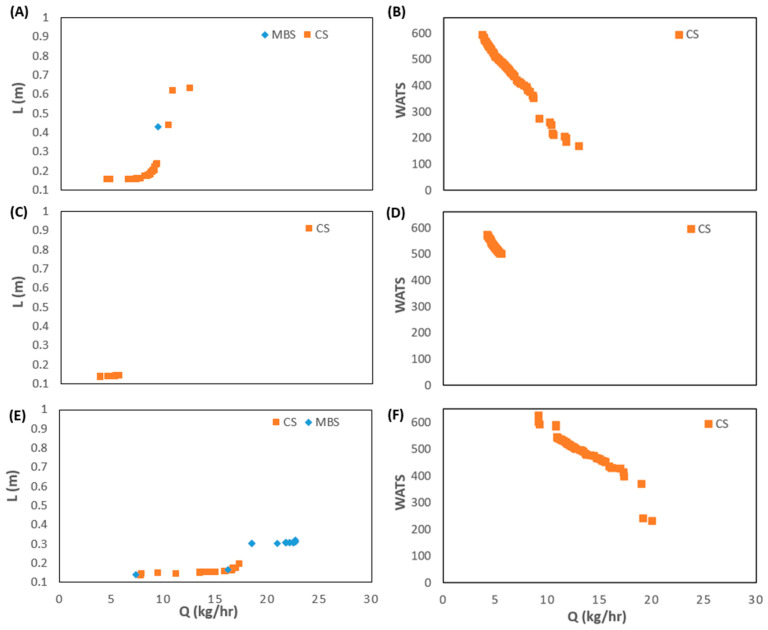
Bi-objective optimization (length for melting, L, and WATS) for PP as follows: (**A**,**B**) Case 9; (**C**,**D**) Case 10; (**E**,**F**) Case 11.

**Figure 11 polymers-15-02212-f011:**
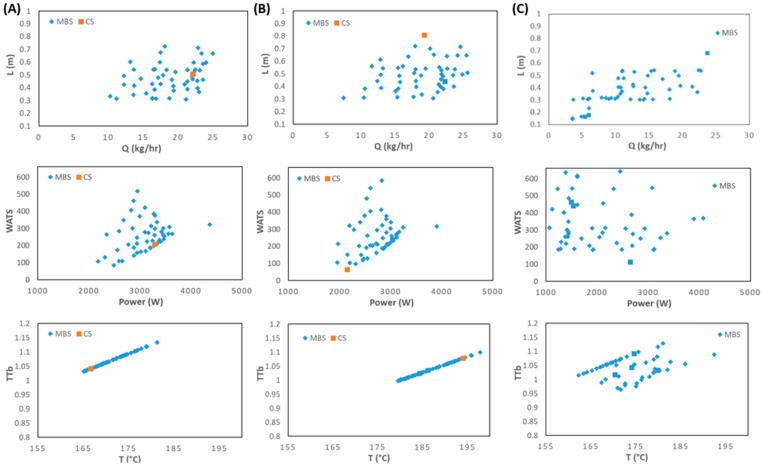
Two-dimensional representation of the optimization results for six objectives for LDPE as follows: (**A**) Case 3; (**B**) Case 6; (**C**) Case 7.

**Figure 12 polymers-15-02212-f012:**
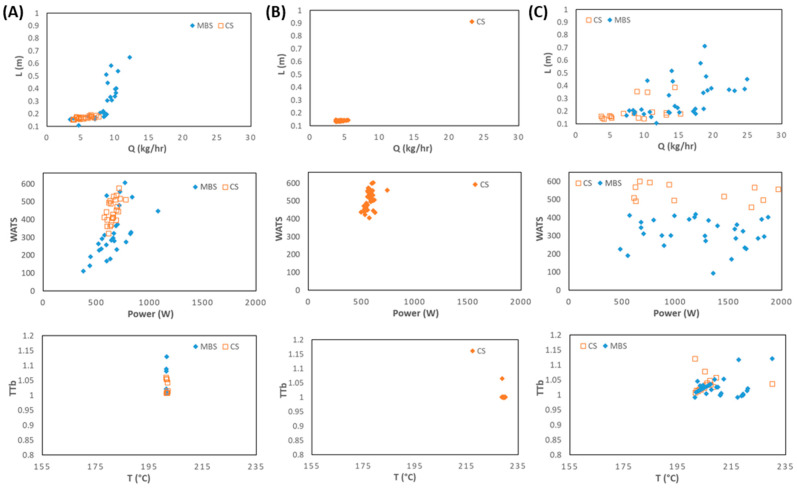
Two-dimensional representation of the optimization results for six objectives for PP: (**A**) Case 8; (**B**) Case 11; (**C**) Case 12.

**Figure 13 polymers-15-02212-f013:**
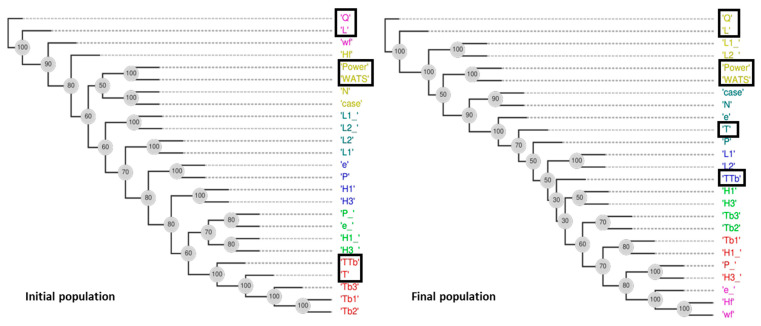
DAMICORE analysis for Case 7 (LDPE) with six objectives.

**Figure 14 polymers-15-02212-f014:**
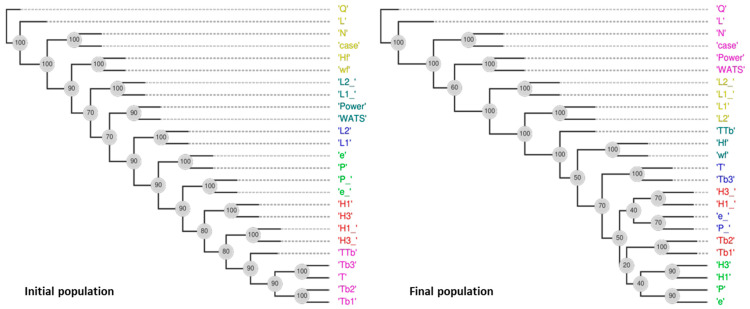
DAMICORE analysis for Case 12 (PP) with six objectives (the colors in the graphs represents the clusters obtained).

**Table 1 polymers-15-02212-t001:** Physical, thermal, and rheological properties of the polymers used in this work as follows: LDPE (LUPOLEN 33FM, LyondellBasell) and PP (ISPLEN 030G1E, Repsol YPF).

Property			LDPE	PP	Units
Density	Solids (ρ_s_)	ρ_∞_	495.0	691	kg/m^3^
	Melt (ρ_m_)	g_0_	923	902	kg/m^3^
Friction coefficients	Internal		0.53	0.50	---
	Hopper		0.30	0.30	---
	Barrel		0.40	0.45	---
	Screw		0.20	0.25	---
Thermal conductivity	Solids	k_s_	0.141	0.210	W/m °C
	Melt	k_m_	0.078	0.180	W/m °C
Specific Heat	Solids (C_s_)	C_0_	2725.0	1882.0	J/kg
	Melt (C_m_)	C_0_	2574.0	1975.0	J/kg
Enthalpy of melting		h	116,100.0	120,490.0	J/kg
Melting temperature		T_m_	100.3	169.1	°C
Viscosity (Carreau-Yasuda law)		η_0_	33,000.0	3041.5	Pa s
		E/R	5000.0	4023.3	K
		λ	1.00	0.17	S
		a	1.80	1.82	---
		n	0.35	0.35	---
		*T* _0_	423.15	533.15	K

**Table 2 polymers-15-02212-t002:** Optimization objectives, aim of the optimization, and allowed range of variation (values for PP between brackets).

Objectives	Aim	*x_min_*	*x_max_*
Output—Q (kg/h)	Maximize	1	30
Length for melting—L (m)	Minimize	0.1	0.9
Melt temperature at die exit—T (°C)	Minimize	150 (LDPE)	210 (LDPE)
Melt temperature at die exit—T (°C)	Minimize	190 (PP)	240 (PP)
Power consumption—Power (W)	Minimize	0	9200
WATS	Maximize	0	1300
Viscous dissipation—TTb	Minimize	0.9	1.2

**Table 3 polymers-15-02212-t003:** Case studies for LDPE (the operating conditions are included as decision variables in Case 7).

Case	Operating Conditions	Decision Variables			
		N (rpm)	Tb1 (°C)	Tb2 (°C)	Tb3 (°C)
1	Constant	40	140	140	140
2	Constant	60	140	140	140
3	Constant	80	140	140	140
4	Constant	40	180	180	180
5	Constant	60	180	180	180
6	Constant	80	180	180	180
7	Variable	[40–80]	[140–180]	[140–180]	[140–180]

**Table 4 polymers-15-02212-t004:** Case studies for PP (the operating conditions are included as decision variables in Case 12).

Case	Operating Conditions	Decision Variables			
		N (rpm)	Tb1 (°C)	Tb2 (°C)	Tb3 (°C)
8	Constant	40	200	200	200
9	Constant	40	230	230	230
10	Constant	70	230	230	230
11	Constant	100	230	230	230
12	Variable	[40–100]	[200–230]	[200–230]	[200–230]

**Table 5 polymers-15-02212-t005:** Geometrical parameters of both CSs and MBSs.

Screw Type	Decision Variables								
CS	case	L1	L2	D1	D3	P	e		
MBS		L1_	L2_	D1_	D3_	P_	e_	Hf	wf
Range of variation	[0,1]	[190–400]	[190–400]	[18–22]	[22–26]	[25–35]	[3–4]	[0.3–0.9]	[2–4]

**Table 6 polymers-15-02212-t006:** Distances between DVs and objectives for Case 7 (LDPE).

	‘Q’	‘L’	‘T’	‘Power’	‘WATS’	‘TTb’	Average
**‘L2_’**	0.21	0.21	0.36	0.28	0.28	0.56	0.32
**‘case’**	0.36	0.36	0.21	0.28	0.28	0.43	0.32
**‘N’**	0.36	0.36	0.21	0.28	0.28	0.43	0.32
**‘L1_’**	0.21	0.21	0.36	0.28	0.28	0.56	0.32
**‘e’**	0.5	0.5	0.21	0.43	0.43	0.28	0.39
**‘P’**	0.5	0.5	0.21	0.43	0.43	0.28	0.39
**‘L2′**	0.56	0.56	0.28	0.5	0.5	0.21	0.43
**‘L1′**	0.56	0.56	0.28	0.5	0.5	0.21	0.43
**‘H1′**	0.71	0.71	0.43	0.64	0.64	0.21	0.55
**‘H3′**	0.71	0.71	0.43	0.64	0.64	0.21	0.55
**‘Tb1′**	0.86	0.86	0.56	0.79	0.79	0.36	0.7
**‘H1_’**	0.86	0.86	0.56	0.79	0.79	0.36	0.7
**‘Tb3′**	0.86	0.86	0.56	0.79	0.79	0.36	0.7
**‘Tb2′**	0.86	0.86	0.56	0.79	0.79	0.36	0.7
**‘e_’**	0.93	0.93	0.64	0.86	0.86	0.43	0.77
**‘P_’**	0.93	0.93	0.64	0.86	0.86	0.43	0.77
**‘H3_’**	0.93	0.93	0.64	0.86	0.86	0.43	0.77
**‘wf’**	1	1	0.71	0.93	0.93	0.5	0.84
**‘Hf’**	1	1	0.71	0.93	0.93	0.5	0.84

**Table 7 polymers-15-02212-t007:** Distances between objectives for Case 7 (LDPE).

	‘Q’	‘L’	‘T’	‘Power’	‘WATS’	‘TTb’	Average
**‘Q’**	0.00	0.07	0.36	0.28	0.28	0.56	0.26
**‘L’**	0.07	0.00	0.36	0.28	0.28	0.56	0.26
**‘T’**	0.36	0.36	0.00	0.28	0.28	0.28	0.26
**‘Power’**	0.28	0.28	0.28	0.00	0.07	0.50	0.24
**‘WATS’**	0.28	0.28	0.28	0.07	0.00	0.50	0.24
**‘TTb’**	0.56	0.56	0.28	0.50	0.50	0.00	0.40

**Table 8 polymers-15-02212-t008:** Best solutions for Case 7 (LDPE).

Weights	Decision Variables													Objectives						t(x)
	Case	N	Tb1	Tb2	Tb3	L1_	L2_	*H1_*	H3_	*P_*	*e_*	*Hf*	*wf*	Q	*L*	T	*Power*	WATS	TTb	
**(0.25; 0.25; 0.25; 0.25)**	61.2	54.2	165	164	175	268.0	170.0	21.7	22.0	34.8	3.1	0.8	3.5	5.5	0.493	189	2358	383	1.5	0.8
	61.2	77.4	165	164	175	267.9	170.0	21.7	22.0	35.0	3.0	0.8	3.5	5.3	0.487	189	2278	308	1.5	0.8
	61.3	63.5	165	160	160	170.8	170.0	22.0	26.0	25.0	3.0	0.8	3.4	5.8	0.517	190	2395	376	1.5	0.8
	59.2	54.1	164	161	160	222.1	170.0	22.0	22.0	35.0	3.0	0.5	3.5	5.7	0.503	190	2397	308	1.4	0.8
	61.5	40.0	165	160	167	170.3	170.0	22.0	26.0	25.0	3.0	0.8	3.4	6.4	0.527	192	2834	316	1.4	0.8
**(0.5; 0.167; 0.167; 0.167)**	61.3	63.5	165	160	160	170.8	170.0	22.0	26.0	25.0	3.0	0.8	3.4	6.1	0.520	191	2583	340	1.4	0.8
	61.4	40.2	162	164	178	263.9	170.0	22.0	22.0	35.0	3.0	0.8	3.4	6.4	0.527	192	2834	316	1.4	0.8
	61.2	54.2	165	164	175	268.0	170.0	21.7	22.0	34.8	3.1	0.8	3.5	5.8	0.517	190	2395	376	1.5	0.9
	59.2	53.8	164	160	160	170.0	170.0	22.0	22.0	35.0	3.0	0.5	3.5	5.7	0.503	190	2397	308	1.4	0.9
	59.1	79.1	164	161	176	221.9	170.0	22.0	22.0	35.0	3.0	0.4	3.5	5.5	0.493	189	2331	328	1.5	0.9

**Table 9 polymers-15-02212-t009:** Distances between DVs and objectives for Case 12 (PP).

	‘Q’	‘L’	‘T’	‘Power’	‘WATS’	‘TTb’	Average
**‘case’**	0.21	0.21	0.64	0.28	0.28	0.43	0.34
**‘L2_’**	0.36	0.36	0.50	0.28	0.28	0.28	0.34
**‘L1_’**	0.36	0.36	0.50	0.28	0.28	0.28	0.34
**‘N’**	0.21	0.21	0.64	0.28	0.28	0.43	0.34
**‘L2′**	0.43	0.43	0.43	0.36	0.36	0.21	0.36
**‘L1′**	0.43	0.43	0.43	0.36	0.36	0.21	0.36
**‘wf’**	0.56	0.56	0.28	0.50	0.50	0.21	0.43
**‘Hf’**	0.56	0.56	0.28	0.50	0.50	0.21	0.43
**‘Tb3′**	0.64	0.64	0.07	0.56	0.56	0.28	0.46
**‘e_’**	0.71	0.71	0.28	0.64	0.64	0.36	0.55
**‘P_’**	0.71	0.71	0.28	0.64	0.64	0.36	0.55
**‘H3_’**	0.86	0.86	0.43	0.79	0.79	0.50	0.70
**‘H1_’**	0.86	0.86	0.43	0.79	0.79	0.50	0.70
**‘Tb2′**	0.93	0.93	0.50	0.86	0.86	0.56	0.77
**‘Tb1′**	0.93	0.93	0.50	0.86	0.86	0.56	0.77
**‘e’**	1.00	1.00	0.56	0.93	0.93	0.64	0.84
**‘P’**	1.00	1.00	0.56	0.93	0.93	0.64	0.84
**‘H3′**	1.00	1.00	0.56	0.93	0.93	0.64	0.84
**‘H1′**	1.00	1.00	0.56	0.93	0.93	0.64	0.84

**Table 10 polymers-15-02212-t010:** Distances between objectives for Case 12 (PP).

	‘Q’	‘L’	‘T’	‘Power’	‘WATS’	‘TTb’	Average
**‘Q’**	0.00	0.07	0.64	0.28	0.28	0.43	0.28
**‘L’**	0.07	0.00	0.64	0.28	0.28	0.43	0.28
**‘T’**	0.64	0.64	0.00	0.56	0.56	0.28	0.45
**‘Power’**	0.28	0.28	0.56	0.00	0.07	0.36	0.26
**‘WATS’**	0.28	0.28	0.56	0.07	0.00	0.36	0.26
**‘TTb’**	0.43	0.43	0.28	0.36	0.36	0.00	0.31

**Table 11 polymers-15-02212-t011:** Best solutions for Case 12 (PP).

Weights	Decision Variables																			Objectives						t(x)
	Case	N	Tb1	Tb2	Tb3	L1	L2	H1	H3	P	e	L1_	L2_	*H1_*	H3_	*P_*	*e_*	*Hf*	*wf*	Q	*L*	T	*Power*	WATS	TTb	
**(0.20; 0.20; 0.20; 0.20)**	98.1	100.0	227	202	205							179.1	203.2	21.7	23.2	35.0	3.0	0.8	2.9	18.6	0.344	212	7514	329	1.1	0.9
	46.3	98.9	230	202	200	170.0	191.0	22.0	23.7	26.0	3.0									13.2	0.171	209	2212	491	1.1	1.0
	46.8	99.0	201	202	202	173.6	187.8	22.0	23.7	26.1	3.0									13.1	0.189	208	2371	483	1.0	1.0
	41.2	99.5	204	202	200	171.8	170.0	22.0	23.6	30.3	3.0									15.2	0.180	207	2166	442	1.0	1.0
	94.1	99.4	230	210	201							400.0	298.1	22.0	23.3	34.9	3.0	0.8	2.4	17.5	0.179	209	1874	404	1.1	1.0
**(0.40; 0.15; 0.15; 0.15)**	98.1	100.0	227	202	205							179.1	203.2	21.7	23.2	35.0	3.0	0.8	2.9	18.6	0.344	212	7514	329	1.1	0.9
	46.3	98.9	230	202	200	170.0	191.0	22.0	23.7	26.0	3.0									13.2	0.171	209	2212	491	1.1	1.0
	46.8	99.0	201	202	202	173.6	187.8	22.0	23.7	26.1	3.0									13.1	0.189	208	2371	483	1.0	1.0
	41.2	99.5	204	202	200	171.8	170.0	22.0	23.6	30.3	3.0									15.2	0.180	207	2166	442	1.0	1.0
	94.1	99.4	230	210	201							400.0	298.1	22.0	23.3	34.9	3.0	0.8	2.4	17.5	0.179	209	1874	404	1.1	1.0
**(0.60; 0.10; 0.10; 0.10)**	98.1	100.0	227	202	205							179.1	203.2	21.7	23.2	35.0	3.0	0.8	2.9	18.6	0.344	212	7514	329	1.1	0.9
	56.6	100.0	200	202	200							193.6	205.0	22.0	22.2	34.0	3.0	0.8	2.3	24.6	0.374	207	1778	285	1.0	1.0
	97.1	99.8	200	200	200							194.9	194.9	21.8	22.5	34.8	3.0	0.8	2.9	23.1	0.360	207	1837	296	1.0	1.0
	83.3	97.6	201	206	200							293.2	210.6	19.4	22.6	34.6	3.0	0.9	2.4	18.6	0.220	207	2008	296	1.0	1.0
	94.1	99.4	230	210	201							400.0	298.1	22.0	23.3	34.9	3.0	0.8	2.4	17.5	0.179	209	1874	404	1.1	1.0

## Data Availability

Not applicable.

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
