# Peer review of "Evolutionary Multi-Objective Optimization of Extrusion Barrier Screws: Data Mining and Decision Making"

_polymers, 2023, doi:10.3390/polym15092212_

Round 1

Reviewer 1 Report

The article is original and interesting. It deals with the issue of barrier screw design and optimization in the single-screw extrusion process. I have no merit comments on the method of optimization research. However, please let me know why there is a different number of test cases regarding polyethylene (7 cases) and polypropylene (5 cases).

I suggest making the following corrections:

1) page 6, line 236, 'Algorith 1 MOEA': I suggest putting in this place a flowchart, as it is clearer.

2) page 7, line 255: The word 'Smetric' should be written as 'S-Metric'

3) page 9, Table 1: Incorrect unit notation: Density Solids is 'Kg/m3' should be 'kg/m3', Viscosity is 'Pa.s' should be 'Pa∙s'.

4) page 10, Table 2: Melt temperature at die exit should be written on two lines, since it applies to two different materials. In brackets, the names of these materials can be given, e.g. 150 (LDPE).

5) page 10 and 11, Table 3: There are two tables with the same number, with the order of cases continuing. I think that the numbering of tables and cases should be separated, because it applies to different materials. The 'Geometry' column is unnecessary. Information about geometry can be included in the table description. On page 11 in the table description there is '... in case 7).' should be '... in case 12).'.

6) page 11, Table 4: I suggest that the decison variables should be clearly identified. For the CS-type screw should be described as e.g., L1CS and for MBS-type screw as L1MBS. This also applies to the content of the article, Table 5, 7, 9, 10 and Figures 13, 14. The range of variation is incorrectly presented e.g., is [190-,400] should be [190-400].

7) page 11, line 409: ' The results...' the sentence is not clear, it suggests that in case 3 the value of N=80 rpm and Tbi=180⁰C, while in table 3 the values N=80 rpm but Tbi=140⁰C.

The paper is very well written, but please double check for minor errors.  For example: page 6, line 236 is 'Algorith 1 MOEA' should be 'Algorithm 1 MOEA'; page 16 line 461 is '5.1- Analyis ...' should be '5.1- Analysis ...'.

Author Response

Reviewer 1

We thank the reviewer for his/her positive remarks and for the valuable comments that helped to improve the paper.

The article is original and interesting. It deals with the issue of barrier screw design and optimization in the single-screw extrusion process. I have no merit comments on the method of optimization research. However, please let me know why there is a different number of test cases regarding polyethylene (7 cases) and polypropylene (5 cases).

ANSWER:

The seven test cases for polyethylene were used to test the effect of different operating conditions, before performing a global optimization also using the operating conditions as decision variables. The same approach was to be applied to polypropylene. However, due to its thermal properties (namely thermal conductivity and melting heat), melting is very fast. Therefore, it was not necessary to perform optimizations for all the test cases.

I suggest making the following corrections:

1) page 6, line 236, 'Algorith 1 MOEA': I suggest putting in this place a flowchart, as it is clearer.

ANSWER:

Since Algorithm 1 is very simple, we are not entirely sure that a flowchart would improve its clarity. Anyway, we hope that the reviewer will agree that numbering the steps makes the algorithm clearer.

2) page 7, line 255: The word 'Smetric' should be written as 'S-Metric'

ANSWER:

Thank you, done.

3) page 9, Table 1: Incorrect unit notation: Density Solids is 'Kg/m3' should be 'kg/m3', Viscosity is 'Pa.s' should be 'Pa∙s'.

ANSWER:

Thank you, corrected.

4) page 10, Table 2: Melt temperature at die exit should be written on two lines, since it applies to two different materials. In brackets, the names of these materials can be given, e.g. 150 (LDPE).

ANSWER:

Thank you, done

5) page 10 and 11, Table 3: There are two tables with the same number, with the order of cases continuing. I think that the numbering of tables and cases should be separated, because it applies to different materials. The 'Geometry' column is unnecessary. Information about geometry can be included in the table description. On page 11 in the table description there is '... in case 7).' should be '... in case 12).'.

ANSWER:

We apologize for mistakenly giving the same number to two tables. We implemented all the corrections suggested.

6) page 11, Table 4: I suggest that the decison variables should be clearly identified. For the CS-type screw should be described as e.g., L1CS and for MBS-type screw as L1MBS. This also applies to the content of the article, Table 5, 7, 9, 10 and Figures 13, 14. The range of variation is incorrectly presented e.g., is [190-,400] should be [190-400].

ANSWER:

All the variables for the MBS have an underscore after the ID, i.e., L1 for the CS and L1_ for the MBS. The sign in the range of variation was corrected.

7) page 11, line 409: ' The results...' the sentence is not clear, it suggests that in case 3 the value of N=80 rpm and Tbi=180⁰C, while in table 3 the values N=80 rpm but Tbi=140⁰C.

ANSWER:

Thank you for pinpointing this. The correct value is for case 6. Thus, we corrected the number of this case to 6: “The results are very similar to those of case 6, when N and Tbi are fixed at 80 rpm and 180° C.”

Comments on the Quality of English Language

The paper is very well written, but please double check for minor errors.  For example: page 6, line 236 is 'Algorith 1 MOEA' should be 'Algorithm 1 MOEA'; page 16 line 461 is '5.1- Analyis ...' should be '5.1- Analysis ...'.

ANSWER:

We checked again the English Language.

Reviewer 2 Report

The manuscript “Evolutionary Multi-Objective Optimization of Extrusion Barrier Screws: Data Mining and Decision Making” by António Gaspar-Cunha, Paulo Costa, Alexandre Delbem, Francisco Monaco, Maria José Ferreira and José Covas presents the results of extrusion parameters optimization based on artificial intelligence algorithms.

It is well-known that suboptimal extrusion parameters can lead to a sufficient degradation of the polymer during extrusion and thus result in poor mechanical properties of the plastic part. Thus it is of high importance to find these optimal sets of extrusion parameters in the scope of efficient processing of the polymers, the durability of the plastic parts and minimization of thermo-mechanical degradation. However, optimization of the extrusion parameters is often carried out through trial and error as the process is complicated and all the parameters are interconnected with each other. Consequently, only a sophisticated mathematical algorithm can tackle these types of problems.

In the present manuscript, the authors addressed the optimization of the extrusion of polypropylene and polyethylene through artificial intelligence algorithms. They carried out multi-parameter optimization for the process. They selected 12 cases for optimization.

I find this manuscript well-designed and well-written. The authors invested significant effort to describe the state-of-the-art in the modeling of extrusion processes, they approached thoroughly the description of the methods they used, and they presented the result in a clear way.

Due to the complexity of the problem, the authors add a lot of optimization results graphs and tables into the manuscript. It makes it slightly hard to get through the text and comprehend the meaning of the paper. As a point of improvement, I would advise the authors to leave 1 polyethylene and one polypropylene case in the main text of the manuscript and to move the rest to the supporting information.

I find this manuscript ready for publication after a minor revision.

Some minor corrections of the misprints are needed. 

Author Response

ANSWER:

We thank the reviewer for the positive comments to our work.

Concerning the suggestion of moving most of the data to the supporting information, we respectfully hope that the reviewer does not mind if all the results remain in the main text, because we believe that they are all important for the discussion and do not increase significantly the length of the manuscript.

Round 2

Reviewer 1 Report

Necessary corrections have been made.